# Rice auxin influx carrier *OsAUX1* facilitates root hair elongation in response to low external phosphate

Jitender Giri [1,2], Rahul Bhosale [1], Guoqiang Huang [1,3], Bipin K. Pandey [1,2], Helen Parker[1], Susan Zappala[1], Jing Yang [1,3], Anne Dievart[4], Charlotte Bureau[4], Karin Ljung [5], Adam Price[6], Terry Rose[7,13], Antoine Larrieu [1], Stefan Mairhofer[1,8], Craig J. Sturrock [1], Philip White[9], Lionel Dupuy[9], Malcolm Hawkesford [10], Christophe Perin[4], Wanqi Liang[1,3], Benjamin Peret[1], Charlie T. Hodgman [1], Jonathan Lynch [1,11], Matthias Wissuwa[7], Dabing Zhang [3,12], Tony Pridmore[1,8], Sacha J. Mooney[1], Emmanuel Guiderdoni[4], Ranjan Swarup[1] & Malcolm J. Bennett [1]

Root traits such as root angle and hair length influence resource acquisition particularly for immobile nutrients like phosphorus (P). Here, we attempted to modify root angle in rice by disrupting the *OsAUX1* auxin influx transporter gene in an effort to improve rice P acquisition efficiency. We show by X-ray microCT imaging that root angle is altered in the *osaux1* mutant, causing preferential foraging in the top soil where P normally accumulates, yet surprisingly, P acquisition efficiency does not improve. Through closer investigation, we reveal that *OsAUX1* also promotes root hair elongation in response to P limitation. Reporter studies reveal that auxin response increases in the root hair zone in low P environments. We demonstrate that OsAUX1 functions to mobilize auxin from the root apex to the differentiation zone where this signal promotes hair elongation when roots encounter low external P. We conclude that auxin and OsAUX1 play key roles in promoting root foraging for P in rice.

[1] Centre for Plant Integrative Biology (CPIB), School of Biosciences, University of Nottingham, Nottingham LE12 5RD, UK. [2] National Institute of Plant Genome Research (NIPGR), New Delhi, India. [3] State Key Laboratory of Hybrid Rice, Shanghai Jiao Tong University, Shanghai, China. [4] CIRAD, UMR AGAP, F34398 Montpellier, Cedex 5, France. [5] Umeå Plant Science Centre, Department of Forest Genetics and Plant Physiology, Swedish University of Agricultural Sciences, SE-901 83 Umeå, Sweden. [6] Institute of Biological and Environmental Sciences, University of Aberdeen, Aberdeen AB24 2TZ, UK. [7] Japan International Center for Agricultural Sciences (JIRCAS), 1-1 Ohwashi, Tsukuba, Ibaraki 305-8686, Japan. [8] School of Computer Science, University of Nottingham, Jubilee Campus, Nottingham NG8 1BB, UK. [9] Ecological Sciences, The James Hutton Institute, Invergowrie, Dundee DD2 5DA, UK. [10] Rothamsted Research, Harpenden, Hertfordshire AL5 2JQ, UK. [11] Department of Plant Science, The Pennsylvania State University, 102 Tyson Building, University Park, PA 16802, USA. [12] University of Adelaide-SJTU Joint Centre for Agriculture and Health, University of Adelaide, Waite Campus, Urrbrae, SA, Australia. [13] Present address: Southern Cross Plant Science, Southern Cross University, Lismore, NSW 2480, Australia. These authors contributed equally: Jitender Giri, Rahul Bhosale, Guoqiang Huang, Bipin K. Pandey, Helen Parker, Susan Zappala. Correspondence and requests for materials should be addressed to M.J.B. (email: malcolm.bennett@nottingham.ac.uk)

Food security represents a pressing global issue. Crop production has to double by 2050 to keep pace with predictions of global population increasing to 9 billion. This target is even more challenging given the impact of climate change on water availability and the drive to reduce fertilizer inputs to make agriculture environmentally sustainable. In both cases, developing crops with improved water and nutrient uptake efficiency by manipulating root architecture, which critically influences nutrient and water uptake efficiency would provide part of the solution. For example, root angle impacts phosphate acquisition efficiency (PAE) as this nutrient preferentially accumulates in the top soil[1,2].

Very few genes that regulate root architecture traits such as root angle have been identified in crop plants to date[3]. In contrast, major progress has been made characterizing genes and molecular mechanisms controlling root angle in the model plant *Arabidopsis thaliana*[4]. AUX1 was one of the first genes identified in *Arabidopsis* to control root angle[5,6] and later shown to encode an auxin influx carrier[7,8]. AUX1 regulates root angle by transporting auxin from gravity-sensing columella cells at the root tip via the lateral root cap to elongating epidermal cells that undergo differential growth to trigger root bending[9,10]. Such detailed functional information in model organisms opens possibilities to perform translational studies to manipulate equivalent root traits in crops controlled by orthologous genes.

In this study, we describe how a translational approach was initially adopted to improve PAE in rice by genetically manipulating the orthologous AUX1 sequence. Reverse genetic studies in rice combined with non-invasive X-ray (microCT) imaging in soil confirmed that root angle was significantly altered in *osaux1* compared to wild-type plants. Nevertheless, physiological experiments performed on *osaux1* (versus wild-type) failed to demonstrate improvement in PAE, suggesting that *OsAUX1* controls other traits important to P acquisition. Further studies revealed *OsAUX1* was also required for rice root hair elongation, an important adaptive response designed to forage for immobile nutrients such as P in the soil[11]. Auxin quantification and reporter lines revealed that under low P conditions, auxin levels are elevated in the root hair zone. We conclude that in response to low external P supply, *OsAUX1* is required to transport elevated auxin from the root apex to the differentiation zone to promote root hair elongation and hence facilitate rice P acquisition. In parallel papers, we demonstrate that this auxin-dependent root hair response to low external P is highly conserved in the dicotyledonous model *Arabidopsis thaliana*[12] and which relies on AUX1 to promote hair elongation via intracellular auxin and calcium signaling[13].

## Results

**Rice root angle is altered by disrupting the *OsAUX1* gene.** The *AUX1* gene family in rice is encoded by five closely related *OsAUX1/LAX* genes (Supplementary Figure 1a). Bioinformatic analysis revealed that the two rice sequences (*Os01g63770* and *Os05g37470*) were closely related to AUX1. In order to identify which rice sequence(s) represents an orthologous gene, we tested the ability of each of their cDNA sequences to complement the *Arabidopsis aux1* agravitropic phenotype. This genetic assay revealed that only one of the *OsAUX1* sequences (*Os01g63770*) was able to successfully rescue the *aux1* mutant's root agravitropic defect (Supplementary Figure 1b, c). Our observations are consistent with previous complementation experiments using *Arabidopsis AUX/LAX* sequences, which revealed that gene family members had undergone a process of sub-functionalization[14].

To test the in planta function of *OsAUX1* in rice directly, we characterized two independent T-DNA insertion lines (3A-51110

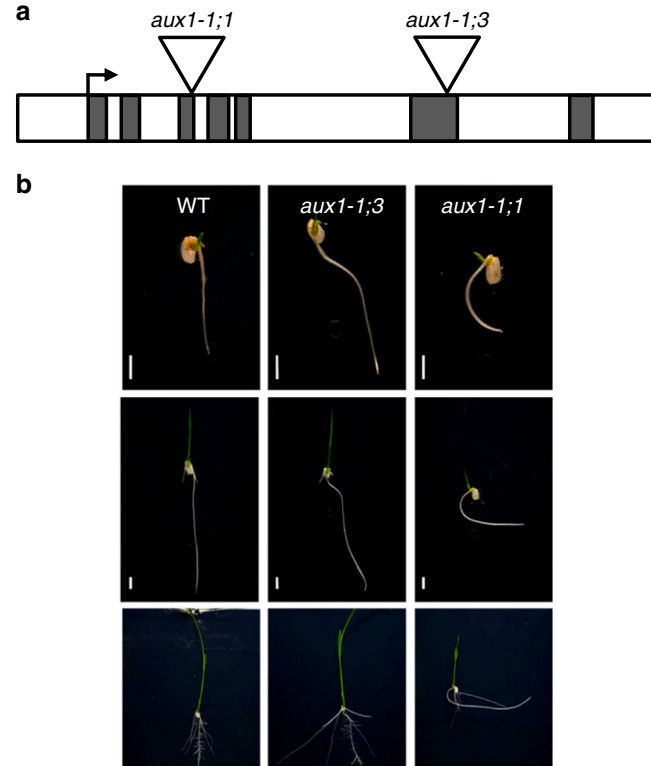

**Fig. 1** *OsAUX1* controls rice root angle. **a** Schematic representation of T-DNA insertion sites in OsAUX1 gene. **b** Time course images of root angle in WT, os*aux1-1;1* and *osaux1-1;3* T-DNA mutants. Images were taken after 3 days after seed germination (3DAG) to 8 days post germination (8DAG). White bars represent 0.5 cm

and 3A-01770) disrupting the *Os01g63770* genomic sequence in the Dongjin background (see "Methods"). The T-DNA insertion lines were termed *osaux1-1;1* and *osaux1-1;3* (in agreement with Zhao et al.[15]). Southern hybridization confirmed that single T-DNA insertion events had disrupted the *OsAUX1* gene in *osaux1-1;1* and *osaux1-1;3*, respectively. PCR amplification of genomic fragments adjacent to each T-DNA followed by sequencing confirmed that T-DNA insertions in *osaux1-1;1* and *osaux1-1;3* had disrupted the gene coding sequence in exon 3 and exon 6, respectively (Fig. 1a). Reverse transcription quantitative-PCR (RT-qPCR) analysis also revealed that both T-DNA alleles exhibited significantly reduced *OsAUX1* transcript abundance (>80%; Supplementary Figure 2). Hence, *osaux1-1;1* and *osaux1-1;3* appear to represent null alleles.

Phenotypic analysis of young seedlings (homozygous for the T-DNA inserts) germinated on vertical agar plates revealed a reduced root angle phenotype in both *osaux1-1;1* and *osaux1-1;3* alleles compared to the positive gravitropic behavior of the wild-type control roots (Fig. 1b). The gravitropic defect became apparent in both primary and crown roots of *osaux1* seedlings 4–8 days after germination (Fig. 1b). Mutant seedling primary and crown roots exhibited altered root angles compared to wild-type roots that grew closer to the vertical (Supplementary Figure 3). Similarly, seedling primary roots of both *osaux1* alleles failed to reorient after a 90° gravity stimulus in contrast to wild-type roots (Supplementary Figure 4). Hence, the *OsAUX1* gene appears to control primary and crown root gravitropic responses and angle in rice.

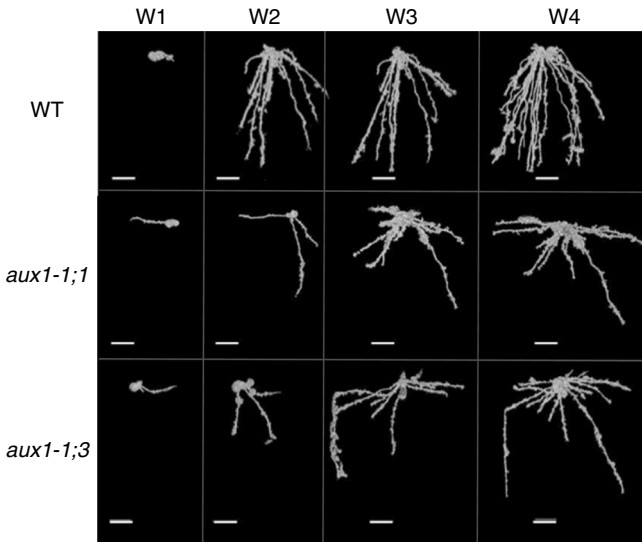

**Fig. 2** MicroCT imaging reveals *OsAUX1* controls root angle in soil. Comparison of root angles from X-ray CT images of soil grown wild-type (WT), os*aux1-1;1* and os*aux1-1;3* roots at 1-, 2-, 3-, and 4-week-old stages (denoted W1–4). Scale bar represents 2 cm

**Phosphorus acquisition efficiency is not improved in *osaux1*.** Root angle represents an important determinant for PAE. Many crops with roots whose angles deviate more from the vertical exhibit greater P foraging ability since this nutrient preferentially accumulates in the upper soil volume[11]. We initially investigated whether *OsAUX1* controls root angle in rice plants grown in soil. The architecture of wild type versus *osaux1* lines was compared using X-ray microCT and rhizotron-based root phenotyping approaches[16]. When using microCT, rice lines were grown in soil for a total of 4 weeks, non-invasively scanning samples every week. This non-destructive imaging approach helped reveal the temporal evolution of wild-type and mutant rice root architecture. Clear differences in root distribution within the soil volume were apparent at week 2 (Fig. 2) with *osaux1* lines preferentially colonizing the upper soil space compared to wild type. Large rhizotrons (1.5 M depth by 0.5 M width) enabled imaging of 2D root architecture in older rice plants, and independently validated differences observed using microCT in root angle and colonization of the upper soil profile by *osaux1* mutant roots (Supplementary Figure 5). Hence, rice plants lacking *OsAUX1* exhibit a major change in the vertical distribution of roots.

Given the striking difference in *osaux1-1* root angle compared to wild type when grown in soil (Fig. 2 and Supplementary Figure 5), we next tested whether the mutant also had improved PAE. We performed a series of experiments designed to assess whether the *osaux1* mutant's root angle phenotype conferred a selective advantage for P foraging. When plants were provided with limited, sufficient and high levels of this immobile nutrient in the soil, no significant difference was evident in P accumulation in shoot tissues of *osaux1* compared to the wild-type control (Supplementary Figure 6). Rather surprisingly, split nutrient treatments (where sufficient or high P were provided in the top 50% soil volume) revealed that *osaux1* accumulated less P in shoot tissue compared to the wild type (Supplementary Figure 6). We conclude, based on the latter observations, that *OsAUX1* must also control other root traits of importance for soil P acquisition.

**OsAUX1 promotes root hair growth in low phosphate conditions.** Root hairs play an important role in accessing immobile nutrients like P from the soil. We therefore examined whether

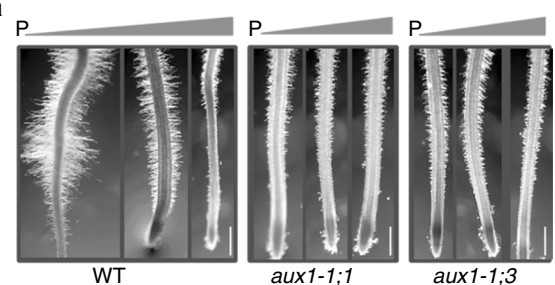

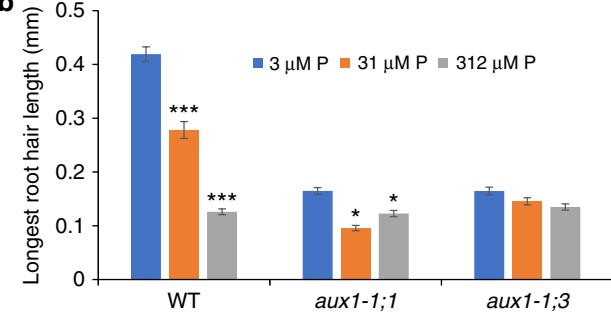

**Fig. 3** *OsAUX1* promotes root hair growth at low external P levels. **a** 9-day-old WT, os*aux1-1;1* and os*aux1-1;3* seedlings were grown for 6 days in hydroponics at three different P concentrations. Scale bar 1 mm. **b** Quantitation of RH length in WT, os*aux1-1;1* and os*aux1-1;3* mutants reveal low P. Each bar represents the average length of 30–60 fully elongated RH on >10 nodal roots. *, **, and *** indicate significant difference *p* value <0.05, 0.001, and 0.0001, respectively. Error bars mean ± SE, *n* = three biological replicates and *p* values were calculated by Student's *t* test

mutating *OsAUX1* disrupted root hair development, in addition to root angle. We initially observed that both *osaux1* mutant alleles retained the ability to form root hairs (Fig. 3a and Supplementary Figure 7). However, closer examination revealed that mutant root hairs were shorter than wild type (Fig. 3b and Supplementary Figure 7). The reduced root hair length in *osaux1* phenocopies the previously reported root hair elongation defect in Arabidopsis *aux1* mutant alleles[17,18] and reveals that this growth response represents a highly conserved AUX1-dependent process.

External phosphate availability has been reported to control root hair length in several plant species[11]. We also observed that external P concentration had a major effect on wild-type rice root hair length (Fig. 3a, b and Supplementary Figure 7), which increased more than threefold to >500 μm under the most limiting nutrient conditions. In contrast, the *osaux1-1;1* and *osaux1-1;3* alleles either exhibited a highly attenuated root hair response or this was completely abolished, respectively (Fig. 3b and Supplementary Figure 7). The marked reduction in root hair length of the *osaux1* alleles (particularly under P limiting conditions) will negatively impact their ability to forage for P in soil. Root hairs account for up to 90% of P uptake[19], and the benefits of increased root length in the top soil profile is more than canceled by the loss of surface area induced by shorter root hairs considering that 91% of the total root surface area is contributed by hairs[20].

**Root auxin response is elevated by low phosphate and OsAUX1.** The observed functional link between OsAUX1 and root hair elongation response to P deficiency suggests roots employ auxin as a signal during this important adaptive response. To directly test whether auxin levels are elevated in rice roots under P limiting conditions, we grew wild-type plants

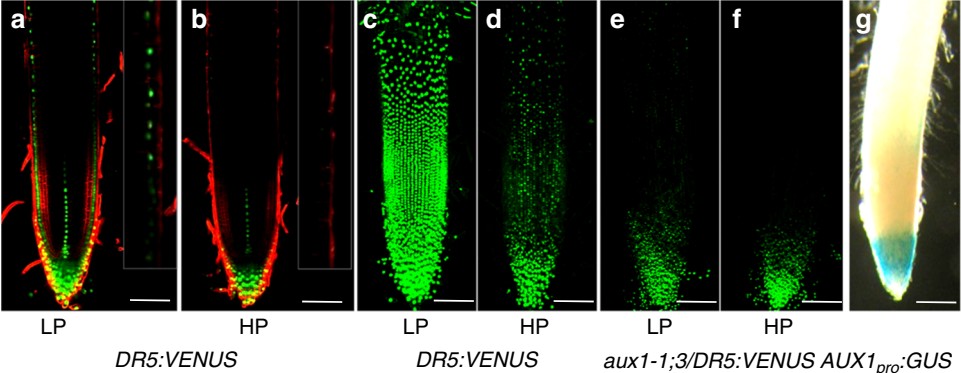

**Fig. 4** Low P increases root hair zone auxin response via AUX1. **a**, **b** Two photon laser scanning microscopy images of auxin response reporter *DR5:VENUS* (green) fluorescence in transgenic rice seedlings grown at either low (**a**) or high P levels (**b**). Inset shows close-up of the distal elongation zone. **c**–**f** Maximum projection confocal images of Z-stacks of *DR5::VENUS* fluorescence in the roots of wild type (**c**, **d**) or *osaux1-1;3* (**e**, **f**) seedlings grown in either low (**c**, **e**) or high P (**d**, **f**). **g** *AUX1pro:GUS* lines reveal *OsAUX1* root apical expression. Scale bar represents 100 μm

hydroponically under low external P supply, then surgically excised root tips and root hair zones and measured levels of the major form of auxin, indole-3-acetic acid (IAA) using GC-MS/MS (see "Methods"). Hormone quantification revealed IAA levels were indeed elevated in wild-type root tip and root hair zone under low external (compared to high) P conditions (Supplementary Figure 8).

To visualize if low external P conditions triggered an auxin response, rice reporter lines encoding the auxin responsive reporter *DR5:VENUSX3* were created (see "Methods"). We monitored changes in rice root auxin response to external P levels employing two forms of laser scanning microscopy (see Methods). Multi-photon microscopy was used to image deep inside rice root tissues, revealing that the *DR5:VENUSX3* reporter signal was elevated in root cap and epidermal cells when grown under low external P (versus high P) conditions (Fig. 4a, b). In parallel, confocal microscopy was employed to image root surface tissues under both external P conditions. A maximal surface projection image was taken to capture the entire cylindrical root surface (Fig. 4c–f). This revealed low *DR5:VENUSX3* auxin response expression in root surface tissues grown in high external P (Fig. 4d), but under low external P conditions reporter activity was strongly upregulated in all root epidermal cells between the apex and hair zone (Fig. 4c).

Lateral root cap and epidermal tissues have been shown in *Arabidopsis* roots to represent the AUX1-mediated conduit for auxin to be transported "shootward" from the root apex to root hair zones[9]. Transgenic rice roots encoding an *OsAUX1* promoter GUS reporter (*OsAUX1:GUS*) revealed that the rice orthologue was expressed in lateral root cap and epidermal tissues (Fig. 4g). To test whether the *osaux1-1;3* mutation reduced auxin-dependent root hair elongation by disrupting "shootward" auxin transport, we monitored *DR5:VENUSX3* reporter expression in the mutant background (Fig. 4e, f). This revealed *DR5:VENUSX3* auxin response expression remained low in root surface tissues grown in either high or low external P. In the latter case, the *DR5:VENUSX3* reporter was clearly elevated in *osaux1-1;3* epidermal cells close to the root apex, but (unlike wild type) was not expressed in more distal cells within the elongation and differentiation zones (Fig. 4e, f). This behavior concurs with model simulations of auxin transport in root tissues, which reveal that influx carrier activity is necessary for this hormone signal to move efficiently from cell to cell[9,10]. We conclude auxin response is elevated in root epidermal cells due to this signal being upregulated at the root apex by low external P, then mobilized to the root hair zone in an OsAUX1-dependent manner.

**Auxin and root hair growth are induced by local phosphate availability**. Given that P is relatively immobile in soil, roots are likely to employ mechanisms to fine tune their hair length in response to this nutrient's heterogeneous distribution. This would necessitate a local (rather than systemic) signaling solution by roots to monitor external P availability and then trigger adaptive responses like hair elongation. To investigate whether root hair length is regulated by either a local or systemic signaling system, rice plants were grown employing a split root experimental set-up, where roots from a single plant were grown in two separate hydroponic chambers to control external P availability. As reported above (Fig. 3), control split roots grown under just low or just high P exhibited long and short root hairs, respectively (Supplementary Figure 9). Interestingly, when roots from individual rice plants were grown simultaneously in high and low external P conditions, they exhibited short and long root hair lengths, respectively (Supplementary Figure 9). Hence, root hair elongation in rice appears to be controlled by local (rather than systemic) P availability. However, when we performed a split plate experiment in soil, where seminal roots from the same rice plant were exposed (at the same time) to replete P and low P conditions, the latter roots exhibited an attenuated hair elongation response compared to control roots (Supplementary Figure 10). This suggests that, while root hair length is strongly influenced by local P availability, a systemic signal(s) may also communicate the P status of shoot tissues.

We next examined whether auxin response plays a role in local and/or systemic signaling mechanisms to P availability using our split root hydroponic system. As reported above, *DR5:VENUSX3* rice split roots grown under just low or just high external P conditions exhibited high and low reporter signals, respectively (Fig. 5a, b and Supplementary Figures 11 and 12). Similarly, when roots from individual rice *DR5:VENUSX3* plants were grown simultaneously in high and low external P conditions, they also exhibited low and high auxin response reporter expression, respectively (Fig. 5a, b and Supplementary Figures 11 and 12). Hence, root auxin response appears to be inversely related to local P availability, where low levels of this key nutrient triggers an increase in root epidermal auxin response, which promotes root hair elongation to better forage for this immobile resource in soil.

## Discussion
Our study has uncovered a novel role for OsAUX1 in facilitating root adaptation to low external P by promoting hair elongation, thereby helping increase the volume of soil being explored by the

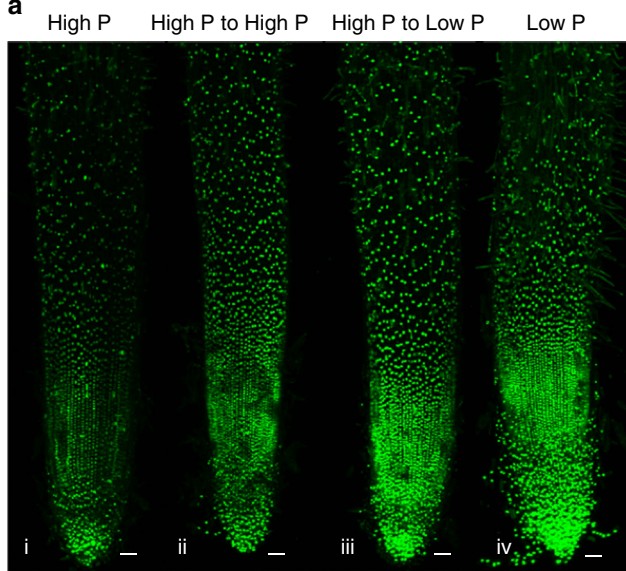

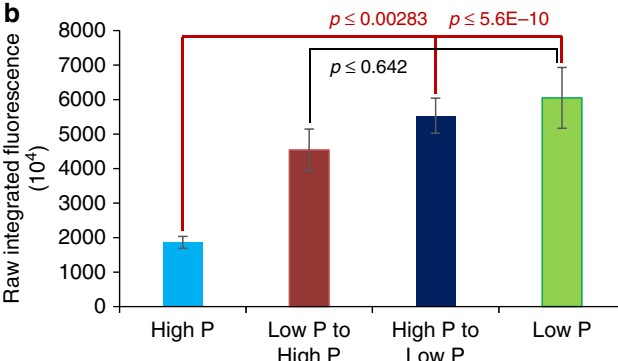

**Fig. 5** Low P root auxin response is independent of plant P status. **a** Maximum projection confocal images of Z-stacks of *DR5::VENUS* fluorescence in the seedlings grown initially in high P medium for 7 days and then transferred to high P (i) for a further 6 days. (ii) and (iii) show *DR5::VENUS* fluorescence of split P experiment roots, where 7-day-old high P roots were split into two halves: one half was grown in high (ii) and the other in low P medium (iii) for a further 6 days. (iv) Maximum projection confocal image of 13-day-old low P grown rice root. **b** Raw integrated fluorescence intensity quantification of *DR5::VENUS* roots (from Fig. 5a and Supplematery Figure 11). Each bar represents the average raw integral density of fluorescence intensity of DR5::VENUS under high P, low P to high P, high P to low P, and low P conditions. Fluorescence intensity of at least 19 roots under low P and high P grown *DR5::VENUS* seedlings and 10 roots of split P conditions were used for fluorescence intensity measurement in three independent replicates. Scale bar represents 50 µm. Student's *t* test was performed to calculate *p* values

plant root. Plant physiologists have long known that low P availability triggers a root hair elongation response in many species[11]. *Arabidopsis* developmental biologists have also observed two decades ago that auxin and AUX1 promote root hair elongation[17,18]. Our current study in rice provides the experimental evidence that integrates these observations and stimulated subsequent efforts in the model plant *Arabidopsis thaliana*[12,13] to develop a mechanistic framework for this adaptive response pathway.

The conservation of the AUX1-regulated root hair adaptive response between model dicot and monocot species provides confidence that we have uncovered a highly conserved auxin regulatory mechanism controlling plant responses to external P

availability. A central role for auxin has been further substantiated by the observation that *Arabidopsis* mutants either disrupting auxin response (e.g., *arf19*), synthesis (e.g., *taa1*), or degradation (e.g., *dao1*) also modify the P deficiency-induced root hair elongation response[12]. In addition, hormone quantification, pharmacological treatment, and reporter studies in rice and *Arabidopsis* have revealed that P-deficit elevates IAA levels and response (Supplementary Figure 8)[12,13], triggering enhanced auxin responsive gene expression in key root tissues that include epidermal root hair cells. Targeting AUX1 to just lateral root cap and epidermal root tissues rescued the *aux1* P deficiency root hair defect, demonstrating the functional importance of the shootward auxin transport pathway from the root apex via the lateral root cap to elongation and differentiation zones[12]. Auxin-inducible transcripts that exhibit elevated expression in the elongation and differentiation zones during P-deficit conditions include the transcriptional factor genes *ARF19* and (its targets) *RSL2* and *RSL4*. Given the recent demonstration that the abundance of RSL4 exhibits a linear relationship with root hair length[21], *RSL4* mRNA upregulation by auxin (in response to P deficit) would promote hair elongation. Collectively, our experimental results can be placed into a mechanistic framework initiated by auxin upregulation at the root apex in response to low external P availability and culminating in upregulation of *RSL2* and *RSL4* in the elongation/differentiation zones that enhances root hair length and P acquisition.

Exactly how low external P availability triggers the upregulation of auxin levels at the root apex has been unclear until now. Split root experiments in rice and Arabidopsis[12] demonstrate that auxin upregulation triggered by low external P was a local (rather than systemic) response. The recent elegant demonstration that P uptake and sensing by a root occurs at the apex[22] raises the intriguing possibility that root cap cells provide a nexus for integrating information about local external nutrient availability that generates physiological signals like auxin. Bhosale et al.[12] have demonstrated that the auxin biosynthesis gene *TAA1* and its protein is upregulated at the root apex in response to low external P levels. As a consequence, elevated auxin levels are transported to other root cells (e.g., epidermal cells) to trigger adaptive responses designed to enhance local root P acquisition (e.g., root hair elongation; Fig. 5). The seventeenth century plant anatomist Grew originally made the connection between plant nutrition and the root tip[23]. The present study establishes how auxin serves as an important signal for P status in the root, linking the root cap and root differentiation zones employing the auxin influx carrier OsAUX1, to promote root hair elongation in order to help capture more P.

## Methods

**Plant material and growth conditions**. *Arabidopsis thaliana* seeds (Col-0) were surface sterilized and grown in a growth room under 16 h light (150–200 µmols m$^{-2}$ s$^{-1}$; 23 °C) and 8 h dark cycle (18 °C). Rice (*Oryza sativa* L. japonica) *AUX1* T-DNA insertion lines *osaux1-1;1* and *osaux1-1;3* (Dongjin background) and Dongjin wild-type seeds were provided by Pr G An, Kyung Hee University, Korea[24]. Rice plants were grown in 13 cm pots (volume 804 cc) filled with a 1:1 (w:w) ratio of John Innes No1 (John Inness, Norwich UK): Levington M3 (JFC Monro, Devon, UK) soil mix, at 28 °C in 12 h light and 12 h dark cycle and regularly irrigated with plant media[25].

**AUX1 complementation experiments**. cDNA sequences for *OsAUX/LAX* genes were PCR amplified from rice root or leaf cDNA libraries, other than *OsLAX1* which was obtained from the rice BAC clone AK111849. Each cDNA was initially cloned into pGEM-T Easy and then the binary vector *pMOGORFLAUX1*[14], which contains the 2 kb promoter region, start codon and the 3'UTR of the *Arabidopsis AUX1* gene. Constructs were then transformed into the Arabidopsis mutant *aux1-22* using the floral-dip method[26]. Primers used for cDNAs amplification are listed in Supplementary Table 1. Root growth and gravitropism analyses were performed on vertical agar plates and quantified as described earlier[27].

**Characterization of osaux1 root architecture**. Two independent T-DNA insertion mutant lines of *Osaux1* were identified using OryGeneDB software[28]. In line 3A-51110, the T-DNA was inserted within intron 5, while in line 3A-01770 the T-DNA was inserted in exon 6 (and termed *osaux1-1;1* and *osaux1-1;3*, respectively). T-DNA insertions were confirmed using the site-finder approach[29]. Root growth and gravitropism analyses were performed on vertical agar plates and quantified as described earlier[27]. Root architecture analysis of soil grown plants was performed using either rhizotrons[30] and X-ray microCT[16]. In the latter case, the germinated seeds were planted in plastic columns containing sandy loam (Newport) soil. For phosphate amendment crushed TSP (44% $P_2O_5$ in 50 mL of deionised water) was mixed thoroughly with the soil. The amended soil was sieved to <2 mm and then packed in polypropylene columns (5.5 cm diameter, 10 cm height, and 0.23 cm thick) to a 1.2 g cm$^{-3}$ density. Columns were microCT scanned at weekly intervals for 4 weeks using a GE NanoTom CT model. Typical settings were 130 kV, 240 μA, 1080 projections, 73 min total scan time, sample-source distance of 22.7 cm, 27.3 μm voxel size with a 0.1 mm copper filter. The relatively long scan time (73 min) was used to obtain the best quality X-ray CT images for the sample size. Each sample received an approximate X-ray dose of 5.9 Gy over the four scans (1.5 Gy each scan) as estimated by the RadPro X-ray Device Dose-Rate Calculator (McGinnis 2002-2009). Root systems were segmented from the X-ray CT generated images using VGStudioMax and measured with VGStudioMax and RooTrak software[31].

**Root hair assays**. Dehusked rice seeds were surface sterilized with 2% bleach and 0.1% Triton for 15 min followed by five washes with sterile water. Seeds were then germinated on moist Whatman paper for 3 days in dark. Uniformly germinated seedlings were then transferred on 1/4th strength MS plates (pH 5.6 with 1% agar) containing 1 μM, 31 μM, or 312 μM P. Low P media were complemented with equimolar concentration of KCl. Seedlings were grown vertically in 12-inch square plates in growth chamber maintained at 28 °C with 12 h of light and 12 h of darkness. After 9 days of growth seedlings were transferred to glass tubes filled with same media without agar (hydroponic system). Liquid media was changed every day and root hair growth was recorded on nodal roots of 15 days old seedlings using a Zeiss stereo zoom microscope (optical zoom ×2.5, digital zoom ×1.2). Experiments were repeated three times. RH length was measured as the average of 30–60 fully elongated root hairs from one seedling. Data from >10 seedlings were used to calculate the final RH length.

**Split root experiments**. Rice seeds (DR5:VENUS3X) were dehusked and were cut into halves to retain only embryo portions (onwards referred as seeds). Seeds were then surface sterilized with 50% bleach for 10 min followed by 10 washes with sterile water. After washing, seeds were dried on sterile Whatman paper for 10 min. Seeds were germinated for 3 days on vertical ½ MS (Murashige and Skoog) plates (supplemented with 0.5% phytagel) in a growth chamber maintained at 28 °C (250–300 μM photons/m²/s). Uniformly germinated seedlings were then transferred to hydroponic solutions of modified Yoshida medium[24] containing 1 μM (low) P in phytotron growth chamber (16 h day (30 °C)/8 h night (30 °C) photoperiod, 250–300 μM photons/m2/s photon density and~ 70% relative humidity). After 7 days of growth in low P (1 μM), 10 low Pi-starved seedlings were split into two glass tubes filled with low (1 μM) and high (312 μM) P Yoshida medium. The liquid medium was changed every day and fluorescence images and Z-stacks were recorded on nodal roots of 13 days old seedlings using Leica SP5 confocal microcope. All recorded images and Z-stacks were processed in Fiji to generate maximal surface projection images and to measure raw integrated densities of fluorescence. The.lif file format was opened in Fiji and all z slices were summed and duplicated. The duplicated image was used for thresholding to visualize the maximum fluorescence pixels. After thresholding, each fluorescence pixel was selected using the ROI manager tool and a ROI number added to that image. Finally, raw integral densities were calculated using the measurement tool.

**Auxin and P measurements in rice plants**. Root tip (~1.5 mm) and differentiation zone (next 2 mm region) from 15 days old rice seedling grown under low and high P were excised under a dissecting stereo microscope and frozen immediately in liquid nitrogen. Twelve–fifteen roots were used per sample with four biological replicates. Five-hundred picograms of $^{13}C_6$-IAA internal standard was added to each sample before purification. Auxin quantification was performed using GC-MS/MS as described earlier[32] with minor modifications. P levels in shoot tissues were measured using ICP-MS.

**Generation of rice reporter lines**. The DR5$_{rev}$::VENUS fragment was composed of a generic synthetic promoter with nine repeats of the auxin response element (AuxRE) motif (TGTCTC) linked to minimal 35S CaMV promoter[33,34], driving the expression of three copies of the YFP VENUS sequence with the nuclear localization signal N7 from maize[35]. The construct was inserted into the pMLBART[36] vector to form the DR5rev::3xVENUS construct. The vector was transformed into rice japonica cultivar 9522 calli using *Agrobacterium tumefaciens* strain EHA105[37]. To create the OsAUX1$_{pro}$:GUS construct, 1.8 kbp of the OsAUX1 promoter sequence was PCR amplified and cloned into Gateway binary vector pGWB3, which contains the GUS gene (Supplementary Figure 2). This vector was then transformed into *Agrobacterium*. Rice transformation was carried out as described earlier[38].

**Two photon laser scanning microscopy (TLSM)**. Plant seeds were sterilized in ethanol 70% for 1 min, and then in 40% sodium hypochlorite for 30 min under agitation. Seeds were transferred to ½ strength MS plates (supplemented with half strength vitamins; 0.8% agar; pH 5.8). Plates were kept at an angle of 15% from the vertical in a growth chamber maintained at 25 °C, 60% humidity, and under a 12 h photoperiod for 3 days. Root tips were counter stained with propidium iodide (PI; 10 μg/ml) for 10 min and were then briefly washed with distilled water thrice. Root tips were mounted in low melting agarose (0.5%) and were scanned typically using a two photon laser scanning microscope. The GFP and PI emissions were collected in separate channels with excitation at 836 nm (Chameleon Ultra II) and 1096 nm (Chameleon Compact OPO), respectively, with a gain set at 600 nm using 2 PMT NDD and 2 PMT BiG detectors. All images were processed using Zeiss ZEN software. For images stack, the auto brightness correction was applied. In some cases, roots were scanned using Leica SP5 confocal microcope with 1.5 μm step size for Z-stacks. Maximum projections were generated using Leica SP5 software,

**RT-qPCR and reporter imaging**. qRT-PCR was performed in three biological and four technical replicates per sample. Total RNA (2 μg) was used for cDNA synthesis using transcriptor first-strand cDNA synthesis kit (Roche)[14]. For GUS assays, samples were kept immersed in ice-cold 90% acetone with gentle shaking for 1 h followed by three washes with sodium phosphate buffer pH 7 for 1 h. Tissues were incubated in GUS staining solution for 3 h at 37 °C[14] and images were taken on a Leica microscope using DIC optics.

**Data availability**. The authors declare that all data supporting the findings of this study are available within the manuscript and its supplementary files or are available from the corresponding author on request.

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

## Acknowledgements

This work was supported by the awards from the Biotechnology and Biological Sciences Research Council [grant numbers BB/G023972/1, BB/R013748/1, BB/L026848/1, BB/M018431/1, BB/PO16855/1, BB/M001806/1, BB/P010520/1]; the European Research Council FUTUREROOTS Advanced Investigator grant [grant number 294729]; Leverhulme Trust [grant number RPG-2016-409]; Royal Society [grant number WM130021, NA140281]; Newton International Fellowship (NF140287) and British Council Newton Bhabha (228144076). This work was also supported by funds from the University of Nottingham Future Food Beacon of Excellence Nottingham Research and PhD+ fellowship schemes; the Interuniversity Attraction Poles Program initiated by the Belgian Science Policy Office [P7/29]; the Swedish Governmental Agency for Innovation Systems (VINNOVA), and the Swedish Research Council (V.R.) to K.L. We also thank Roger Granbom (Swedish University of Agricultural Sciences) for technical assistance and Gabriel Castrillo for commenting on the manuscript text. Part of this work has been conducted at the Rice Functional Genomics REFUGE platform funded by Agropolis Fondation in Montpellier, France. We also thank DBT-CREST BT/HRD/03/01/2002.

## Author contributions

J.G., R.B., G.H., B.K.P., H.P., S.Z., J.Y., A.D., C.B., K.L., A.P., T.R., A.L., S.M. and C.J.S. performed experiments and contributed experimental data; J.G., R.B., B.K.P., K.L., T.R., P.W., L.D., M.H., C.P., W.L., B.P., C.T.H., J.L., M.W., D.Z., T.P., S.J.M., E.G., R.S. and M. J.B. designed experiments; and J.G., R.B., G.H., B.K.P., R.S. and M.J.B. wrote the manuscript.

## Additional information

**Competing interests:** The authors declare no competing interests.

