## [Peer Review File · Nature Communications]

Reviewers' comments:

Reviewer #1 (Remarks to the Author):

The results presented in this report are quite similar to those presented in the accompanying paper by Bhosale et al, in terms of the role of auxin transport and AUX1 on root hair development in rice. The results presented in this paper include several findings that were previously reported by other authors.

1) The role of OsAUX1 in primary root elongation and root hair formation were previously reported by Yu et al (The Plant Journal 83: 818-830). The same gene was identified as the rice orthologous of AtAUX1, mutation in which lead to a shorter root phenotype with shorter root hairs. The mutants presented in this paper corresponds to mutants previously characterized and reported as OsAUX1 by Zhao et al (Plant, Cell and environment 38: 2208-2222). The only novel results presented in this results section of the paper is the characterization of the agravitropic phenotype of the *osaux1* mutants which is like that previously reported for Arabidopsis.

2) An interesting part of this report is the characterization of the root architecture of rice *osaux1* mutant seedlings using X-ray microCT. These results show that the root system of *osaux1* seedlings, although less robust than the WT, tend to colonize predominantly the upper layers of the soil in comparison to the WT that tends to have longer roots that grow to deeper layers of the soil. This phenotype is not completely unexpected given the agravitropic nature of the *osaux1* roots. The root architecture phenotype of *osaux1* seedlings has been reported in other species to increase phosphate uptake efficiency, which unfortunately in this case turned out to have no effect on PUE.

3) As previously reported by Yu et al, in this paper it is confirmed that OsAUX1 plays a role in root hair elongation, which is similar to that also reported by other authors for Arabidopsis.

4) As discussed for the case of the Arabidopsis *aux1* mutants, the *osaux1* seedlings present a reduction in root hair length in both high and low Pi, but remain responsive to the stimulus of low Pi. Therefore, the authors confirm again that AUX1 plays a role in root hair elongation that has no direct relationship with the low Pi response, but indirectly affect their phenotype because these seedlings are partially impaired in root hair elongation. Root hair length in *Osaux1-1* and *Osaux1-2* are still 3 to 4 times longer in low Pi than in high Pi.

5) Split root experiments are also presented to show that the root hair elongation response to low Pi are like those previously reported for Arabidopsis.

It is not rare to see that two different research groups publish back to back papers reporting similar findings in the same or different species in a high impact journal. However, I find a bit unusual that two groups under the leadership of the same senior scientists want to publish back to back findings that are quite similar for two different species. One paper corroborates what the other paper reports, with of course some difference, which in my opinion are not very relevant for the main conclusions drawn by the authors in the two papers. Nevertheless, I find that the data reported in this paper, is more of a confirmatory nature of previous reports than presenting truly novel data that could be of the general interest required to be published in Nature Communications. I was tempted to suggest that perhaps merging the two papers a higher quality report could emerge that fulfills the requirements to be published in a high impact journal, however, I think that not even reporting the data from the two papers in a single publication would be sufficiently novel to warrant publication in Nature Communications. As stated for the AUX1 Arabidopsis paper, experiments are well designed and the results are robust and provide compelling evidence confirming the important role of auxin transport and AUX1 in root hair elongation, but fail to provide an insight into the mechanisms that regulates root hair elongation in response to low Pi.

Reviewer #2 (Remarks to the Author):

The rice auxin influx carrier OsAUX1 facilitates root hair elongation in response to low external phosphate by Jitender Giri et al. is one of three manuscripts that have been co-submitted. I am really enthusiastic about the message of the stories together; they each show a different angle of a multi-faceted message that links root positioning and root hair growth with auxin synthesis and directed transport. The studies claim to have identified a major regulatory mechanism that controls traits of agronomic importance. Each of the stories have their own focus and interesting points. It describes a series of elegant experiments that support a clever line of thought.

This manuscript focusses on the adaptive responses of the crop plant rice to low external phosphorus and their regulation by the auxin influx transporter, AUX1. The manuscript conveys mechanistic insight in the regulation of these adaptive responses. This work would be an excellent basis for breeding efforts to improve phosphorus uptake.

After reading the manuscripts, I have the impression that there is a major flaw in the interpretation of the root hair length reduction of root hairs of aux1 mutants that have been grown under low P conditions. In this manuscript the authors state on page 6 of the manuscript that P concentration has a major effect on wild type rice root hair length. It increases 3-fold under the most limiting conditions. The length of aux1 mutant root hairs is strongly reduced under conditions where P is not limiting, which limits P uptake. I can follow this. But now the next step: the authors claim that there is a functional link between OsAUX1 and root hair elongation response to P deficiency. I do not see this in the data: Fig. 3 shows that both wild type and aux1 mutant root hair length increases ~3-fold under low P conditions when compared to high P conditions. Yes, the root hairs of aux1 mutants are shorter than wild type root hairs under both low and high P conditions; yes, the absolute increase in length of aux1 root hairs grown under low P conditions is far lower than the absolute increase in length of wild type root hairs. However, the mutant root hairs appear to increase ~3-fold in length when grown on low P medium when compared to mutant root hairs that are grown on high P medium. To me, this discredits the statement that there is a functional link between OsAUX1 and the root hair elongation response to P deficiency. Unfortunately, the manuscripts all build on this reasoning that appears incorrect to me and a solid link between AUX1 and root hair elongation is missing. Perhaps I am overlooking something relevant; I would be very much interested to read a response to this by the authors.

Besides this major issue, the manuscript would benefit from a quantitative analysis of some of the experiments. Although Fig. 4 looks convincing, I could argue that a slight difference in focal plane or root curvature between A and B is causing the decreased fluorescence in the centre of the roots. Likewise, the differences in fluorescence intensities between Fig C and D could be caused by root curvature or a different working distance between root and objective. I would recommend the authors to quantify their data, provide the number of replicates and use internal controls (for example propidium iodide staining intensities) to correct for, for example, differences in working distance. How were the relative fluorescence intensities determined in the figures 5 and S9? Was segmentation used to discriminate between nuclear signal and background?

Although the gravitropic responses of plants were quantified (Fig S3) , I would also recommend quantification of the data in Figs 1 and 2.

Response to Reviewers' comments on NCOMMS-17-00623:

Reviewer #1 (Remarks to the Author):

The results presented in this report are quite similar to those presented in the accompanying paper by Bhosale et al, in terms of the role of auxin transport and AUX1 on root hair development in rice.

To clarify, our logic behind submitting the manuscripts back to back was our observation that the low P induced root adaptive response of root hair elongation was promoted by a common signal (auxin) in distinct monocot and dicot plant species provided us with much greater confidence of the evolutionary importance of this mechanism. In addition, the experiments performed in each manuscript are largely distinct, reflecting the relative wealth of auxin-related experimental tools available in Arabidopsis versus rice to probe the auxin-related mechanisms underpinning this adaptive response.

>The results presented in this paper include several findings that were previously reported by other authors.

1) The role of OsAUX1 in primary root elongation and root hair formation were previously reported by Yu et al (The Plant Journal 83: 818-830. The same gene was identified as the rice orthologous of AtAUX1, mutation in which lead to a shorter root phenotype with shorter root hairs. The mutants presented in this paper corresponds to mutants previously characterized and reported as OsAUX1 by Zhao et al (Plant, Cell and environment 38: 2208-2222). The only novel results presented in this results section of the paper is the characterization of the agravitropic phenotype of the osaux1 mutants which is like that previously reported for Arabidopsis.

Whilst the papers cited above describe several rice aux1 mutant phenotypes, none of these papers describe a role for OsAUX1 in the low P induced root hair elongation response (which represents the major novel finding reported in our manuscript). For example, in Yu et al, the authors investigate the impact of the shorter root hair phenotype in osaux1 on sensitivity to the toxic element Cadmium, whilst Zhou et al primarily studied the osaux1 mutant's reduced lateral root defect.

2) An interesting part of this report is the characterization of the root architecture of rice osaux1 mutant seedlings using X-ray microCT. These results show that the root system of osaux1 seedlings, although less robust than the WT, tend to colonize predominantly the upper layers of the soil in comparison to the WT that tends to have longer roots that grow to deeper layers of the soil. This phenotype is not completely unexpected given the agravitropic nature of the osaux1 roots. The root architecture phenotype of osaux1 seedlings has been reported in other species to increase phosphate uptake efficiency, which unfortunately in this case turned out to have no effect on PUE.

We appreciate the reviewer's positive comments above about use of novel X-ray CT based root imaging approaches. In addition, the lack of an improved PUE in osaux1 was also a surprise to us, but this observation led us to later discover a role for OsAUX1 in the low P induced root hair elongation response (which was later validated in Arabidopsis as described in the accompanying manuscript by Bhosale et al).

3) As previously reported by Yu et al, in this paper it is confirmed that OsAUX1 plays a role in root hair elongation, which is similar to that also reported by other authors for Arabidopsis.

To re-iterate point 1 above, Yu et al. did not describe a role for OsAUX1 in the low P induced root hair elongation response (which represents the major novel finding reported in our manuscript). Previous papers have described a role for AUX1 in promoting Arabidopsis root hair elongation. However, no previous paper has described a role for Arabidopsis or rice AUX1 in the low P induced root hair adaptive response.

4) As discussed for the case of the Arabidopsis aux1 mutants, the osaux1 seedlings present a reduction in root hair length in both high and low Pi, but remain responsive to the stimulus of low Pi. Therefore, the authors confirm again that AUX1 plays a role in root hair elongation that has no direct relationship with the low Pi response, but indirectly affect their phenotype because these seedlings are partially impaired in root hair elongation. Root hair length in Osaux1-1 and Osaux1-2 are still 3 to 4 times longer in low Pi than in high Pi.

Please note that the original values in Fig 3B were calculated using root hair length measurements collected from the end of the elongation zone to the maturation zone (i.e. as root hairs were still undergoing elongation), then averaged. We appreciate that such values would not represent the final length reached by root hairs. Hence, we re-analysed the same image datasets, measuring 30-60 fully elongated root hairs, which were then averaged. This revealed that fully elongated root hairs in WT and osaux1 exhibit more than 3-fold versus 1.5-fold difference when grown in low P versus high P, respectively. We also repeated the experiment and observed a similar behaviour (see Fig S7). We conclude from our new and re-analysed rice maximal root hair length data that the low P induced root hair response is either severely impaired or abolished in osaux1 roots.

5) Split root experiments are also presented to show that the root hair elongation response to low Pi are like those previously reported for Arabidopsis.

We were unable to find the above Arabidopsis information. In addition, our split root experiment in rice represents the first reported use of an auxin reporter to study P adaptive responses and independently validate the role of this hormone.

It is not rare to see that two different research groups publish back to back papers reporting similar findings in the same or different species in a high impact journal. However, I find a bit unusual that two groups under the leadership of the same senior scientists want to publish back to back findings that are quite similar for two different species.

As noted above, the logic behind submitting the manuscripts back to back was based on our observation that the low P induced root adaptive response of root hair elongation was promoted by a common signal (auxin) in highly divergent monocot and dicot plant species. Hence, this provided us with much greater confidence of the evolutionary importance of this mechanism.

One paper corroborates what the other paper reports, with of course some difference, which in my opinion are not very relevant for the main conclusions drawn by the authors in the two papers. Nevertheless, I find that the data reported in this paper, is more of a confirmatory nature of previous reports than presenting truly novel data that could be of the general interest required to published in Nature Communications.

I want to clarify that the observed lack of an improved PAE in osaux1 led us to discover a role for OsAUX1 in the low P induced root hair elongation response (and this was later validated in Arabidopsis as described in the accompanying manuscript). Hence, the findings made in the rice paper came before the Arabidopsis manuscript, hence we argue strongly against this being considered 'confirmatory'.

I was tempted to suggest that perhaps merging the two papers a higher quality report could emerge that fulfills the requirements to be published in a high impact journal, however, I think that not event reporting the data from the two papers in a single publication would be sufficiently novel to warrant publication in Nature Communications. As stated for the AUX1

Arabidopsis paper, experiments are well designed and the results are robust and provide compelling evidence confirming the important role of auxin transport and AUX1 in root hair elongation, but fail to provide an insight into the mechanisms that regulates root hair elongation in response to low Pi.

We would strongly argue that the experiments performed in each manuscript are largely distinct and novel, such as the use of X-ray CT to characterise a rice root mutant, as noted by reviewer 1 above. The relative wealth of auxin-related experimental tools available in Arabidopsis versus rice obviously enabled us to probe in greater detail the auxin-related mechanisms underpinning this adaptive response. Nevertheless, the novel use of auxin reporters combined with split root experiments enabled us to uncover the importance of local hormone signalling in this P adaptive root response. Moreover, submitting the manuscripts back to back describing the low P induced root adaptive response of root hair elongation being promoted by a common signal (auxin) in highly divergent monocot and dicot plant species provides readers with much greater confidence of the evolutionary importance and significance of this mechanism.

Reviewer #2 (Remarks to the Author):

The rice auxin influx carrier OsAUX1 facilitates root hair elongation in response to low external phosphate by Jitender Giri et al. is one of three manuscripts that have been co-submitted. I am really enthusiastic about the message of the stories together; they each show a different angle of a multi-faceted message that links root positioning and root hair growth with auxin synthesis and directed transport. The studies claim to have identified a major regulatory mechanism that controls traits of agronomic importance. Each of the stories have their own focus and interesting points. It describes a series of elegant experiments that support a clever line of thought.

We greatly appreciate these positive comments about the quality and interest of this (and the other 2) manuscript(s).

This manuscript focusses on the adaptive responses of the crop plant rice to low external phosphorus and their regulation by the auxin influx transporter, AUX1. The manuscript conveys mechanistic insight in the regulation of these adaptive responses. This work would be an excellent basis for breeding efforts to improve phosphorus uptake.

We completely agree with reviewer 2 that lessons learnt from these studies (i.e. do not target key auxin transport components to manipulate root angle) provide important findings for breeders as they move to adopt genome-editing driven approaches.

After reading the manuscripts, I have the impression that there is a major flaw in the interpretation of the root hair length reduction of root hairs of aux1 mutants that have been grown under low P conditions. In this manuscript the authors state on page 6 of the manuscript that P concentration has a major effect on wild type rice root hair length. It increases 3-fold under the most limiting conditions. The length of aux1 mutant root hairs is strongly reduced under conditions where P is not limiting, which limits P uptake. I can follow this. But now the next step: the authors claim that there is a functional link between OsAUX1 and root hair elongation response to P deficiency. I do not see this in the data: Fig. 3 shows that both wild type and aux1 mutant root hair length increases ~3-fold under low P conditions when compared to high P conditions. Yes, the root hairs of aux1 mutants are shorter than wild type root hairs under both low and high P conditions; yes, the absolute increase in length of aux1 root hairs grown under low P conditions is far lower than the absolute increase in length of wild type root hairs. However, the mutant root hairs appear to increase ~3-fold in length when grown on low P medium when compared to mutant root hairs that are grown on high P medium. To me, this discredits the statement that there is a functional link between OsAUX1 and the root hair elongation response to P deficiency.

Unfortunately, the manuscripts all build on this reasoning that appears incorrect to me and a solid link between AUX1 and root hair elongation is missing. Perhaps I am overlooking something relevant; I would be very much interested to read a response to this by the authors.

Following up this important point (also raised by reviewer 1), we checked how the values in Fig 3B were calculated and noted that root hair lengths were measured at different positions from the end of the elongation zone to the maturation zone (i.e. as root hairs were still undergoing elongation), then averaged this value. We appreciate that such values would not represent the final length reached by root hairs. Hence, we re-analysed the existing dataset, measuring instead 30-60 fully elongated root hairs, then averaging this value. This revealed that fully elongated root hairs in WT and *osaux1* exhibit more than fold versus 1.5-fold difference when grown in low P versus high P, respectively. We recently repeated the experiment and observed a similar behaviour. Our new and re-analysed rice maximal root hair length data suggests that the low P induced root hair growth response is severely impaired in *osaux1* roots. We have now included both the re-analysed data plus new experimental data in Fig. 3B and Fig. S7 respectively. We conclude OsAUX1 plays a key role in the low P response. This was independently validated in DR5:VENUS reporter experiments (Fig 4) which revealed the importance of OsAUX1 in mobilising auxin to elongation zone tissues to trigger auxin responses including root hair elongation.

Besides this major issue, the manuscript would benefit from a quantitative analysis of some of the experiments. Although Fig. 4 looks convincing, I could argue that a slight difference in focal plane or root curvature between A and B is causing the decreased fluorescence in the centre of the roots. Likewise, the differences in fluorescence intensities between Fig C and D could be caused by root curvature or a different working distance between root and objective. I would recommend the authors to quantify their data, provide the number of replicates and use internal controls (for example propidium iodide staining intensities) to correct for, for example, differences in working distance. How were the relative fluorescence intensities determined in the figures 5 and S9? Was segmentation used to discriminate between nuclear signal and background?

We appreciate this feedback and have repeated the experiment and quantified our data using 10-15 replicates (see new Fig 5 and S11 and 12 in our revised manuscript). Details about how we calculate the relative fluorescence intensity is provided in the revised materials and methods section.

Although the gravitropic responses of plants were quantified (Fig S3) , I would also recommend quantification of the data in Figs 1 and 2.

We have actioned this recommendation in our revised manuscript.

Reviewers' comments:

Reviewer #1 (Remarks to the Author):

The new version of this paper provides new data and explanations that answer most of my questions and now presents a complete and more interesting story. Results on the role of auxin transport in promoting root foraging for P are certainly novel, as it affects both root system architecture and root hair formation. However, I still have some reserves about the interpretation that OsAUX1 play a key role in the root hair response to low Pi. Auxin is required for root hair elongation and in the Osaux1 mutant the level of auxin might be below that needed to promote root hair elongation and that is why the mutant do not response to low Pi certainly novel for rice, but not because AUX1 is involved in Pi signaling. Nevertheless, the authors have been careful in interpreting their results and now their main conclusions are well supported by the data presented.

NCOMMS-17-00631A-Z (Giri et al)

The rice auxin influx carrier OsAUX1 facilitates root hair elongation in response to low external phosphate

Response to reviewer comments

Reviewer 1

The new version of this paper provides new data and explanations that answer most of my questions and now presents a complete and more interesting story. Results on the role of auxin transport in promoting root foraging for P are certainly novel, as it affects both root system architecture and root hair formation. However, I still have some reserves about the interpretation that OsAUX1 play a key role in the root hair response to low Pi. Auxin is required for root hair elongation and in the *Osaux1* mutant the level of auxin might be below that needed to promote root hair elongation and that is why the mutant do not response to low Pi certainly novel for rice, but not because AUX1 is involved in Pi signaling. Nevertheless, the authors have been careful in interpreting their results and now their main conclusions are well supported by the data presented.

We are very pleased to note the positive comments made by reviewer 1 about the novelty of our work and acknowledgement that our revised manuscript '*now presents a complete and more interesting story.*'

The remaining concern raised by reviewer 1 relates to the role of AUX1 in low P mediated signalling. As we demonstrate in this (and the co-submitted Arabidopsis manuscript by Bhosale et al), AUX1 functions to mobilise auxin synthesised in the root apex in response to low P. AUX1 is crucial for root hair elongation under low P by facilitating the mobilisation of auxin to epidermal cells in the root hair zone. We agree with reviewer 1 that auxin levels are likely to be below the threshold level required to trigger root hair elongation in the *osaux1* mutant (as revealed using the auxin reporter DR5:NLS-3XVENUS in the *osaux1* versus WT background in Fig. 4e versus 4c), and where this threshold is achieved under low P (versus HP) through TAA1 mediated auxin synthesis (demonstrated in the accompanying manuscript by Bhosale et al).

As the reviewer has not raised other concerns/points, the rice manuscript (NCOMMS-17-00631A-Z (Giri et al) is being re-submitted with minor edits (denoted in red).